# Temporal and genetic variation in female aggression after mating

**Eleanor Bath** [ID]**[1]\*, Edmund Ryan Biscocho[1], August Easton-Calabria[2], Stuart Wigby[1]**

**1** Department of Zoology, University of Oxford, Oxford, United Kingdom, **2** Organismic and Evolutionary Biology Department, Harvard University, Cambridge, Massachusetts, United States of America

\* eleanor.bath@zoo.ox.ac.uk

**Data Availability Statement:** Data for all experiments conducted for this paper will be made publicly available on the Oxford University

## Abstract

Aggression between individuals of the same sex is almost ubiquitous across the animal kingdom. Winners of intrasexual contests often garner considerable fitness benefits, through greater access to mates, food, or social dominance. In females, aggression is often tightly linked to reproduction, with females displaying increases in aggressive behavior when mated, gestating or lactating, or when protecting dependent offspring. In the fruit fly, *Drosophila melanogaster*, females spend twice as long fighting over food after mating as when they are virgins. However, it is unknown when this increase in aggression begins or whether it is consistent across genotypes. Here we show that aggression in females increases between 2 to 4 hours after mating and remains elevated for at least a week after a single mating. In addition, this increase in aggression 24 hours after mating is consistent across three diverse genotypes, suggesting this may be a universal response to mating in the species. We also report here the first use of automated tracking and classification software to study female aggression in *Drosophila* and assess its accuracy for this behavior. Dissecting the genetic diversity and temporal patterns of female aggression assists us in better understanding its generality and adaptive function, and will facilitate the identification of its underlying mechanisms.

## Introduction

Aggression towards conspecifics is prevalent throughout the animal kingdom. Winners in aggressive encounters often benefit from increased access to resources or mates, better positions in social hierarchies, and ultimately, higher reproductive success [1–4]. Male aggression has been shown to be at least partly heritable in flies, mice, rats, and chickens, suggesting a genetic basis [5–8]. Although we know a substantial amount about the genetic basis of male aggression, we know little about it in the context of female aggression. As males and females show striking differences in the frequency, form, and intensity of aggression they display, it seems likely that there are also significant sex differences in the genetic architecture underlying intrasexual aggression. As genetic variation forms the pool of variation on which selection can act, determining how much genotypes differ is a key question for understanding how female aggression has evolved.

Research Archive. The doi is 10.5287/bodleian:
NGYZrDjOq.

**Funding:** This study was funded by a Junior
Research Fellowship from Christ Church college,
University of Oxford, and a John Fell Fund grant to
E.B., as well as a BBSRC fellowship to S.W. (BB/
K014544/1). The funders had no role in study
design, data collection and analysis, decision to
publish, or preparation of the manuscript.

**Competing interests:** The authors have declared
that no competing interests exist.

Females display a large degree of plasticity in their aggressive behavior across time, adding another important source of variation—temporal variation. Females typically fight over resources associated with offspring production or survival, and there is often a tight temporal association between female aggression and reproduction [2,9,10]. Classifying these temporal associations between aggression and reproduction in females can help us understand female aggression in a number of ways. First, investigating what stages of reproduction lead to an increase (or decrease) in aggression can help us to establish the adaptive value of aggression for females. As aggression is expected to have costs, in the form of energy expenditure, injury, and opportunity costs, females should only elevate their aggression when the benefits outweigh the costs. If these costs and benefits shift over a female's reproductive cycle, we expect to see plastic expression of aggression with the most elevated levels of aggression displayed when aggression is most beneficial for females. The optimal time for aggression will differ across species, depending heavily on what resources are valuable and when they are most scarce. In a number of species, the benefits are highest when investment in reproduction is at its highest–e.g. after mating, or during gestation and lactation [11–13]. For other species, mates and associations with males are more valuable, leading to higher rates of aggression when females are ready to mate [10,14] Second, dissecting the temporal associations between aggression and reproduction can assist us in pinpointing potential mechanisms that regulate female aggression and whether these differ from those regulating male aggression [15]. By tracking the timing of aggression and any behavioral or physiological correlates, we are able to identify putative mechanisms for further testing.

*Drosophila melanogaster* is a model organism for the study of male aggression and sexual selection [16–18], though fewer studies have examined female aggression (out of 65 empirical studies found on a Web of Science search using the keywords 'Drosophila melanogaster' and 'aggress*', 61 measured male aggression and 8 measured female aggression–search conducted 30.4.19). Females in this species fight over food, particularly valuable protein-rich resources, such as live yeast [19–21]. Mated females will fight for twice as long over access to food as virgin females, and this change is stimulated by the transfer of sperm and seminal fluid proteins during mating [11]. Mating also dramatically alters other aspects of female behavior and physiology—mated females show increased rates of ovulation and egg laying, reduced receptivity to mating attempts, changed feeding patterns and preferences, altered immune responses, and even changes in sleep patterns [22]. However, there is substantial variation amongst populations and genotypes in the strength and speed of these post-mating responses (PMRs). For example, females from the commonly used $w^{1118}$ lab strain eject mating plugs almost twice as fast as females from a Canton S background [23]. Males from different populations also differ in their ability to stimulate female PMRs [24], which suggests differences in male ejaculate composition or co-evolution between males and females that alters the strength of PMRs in different populations. The variation amongst different populations and lab strains in PMRs suggest that mating-induced female aggression may vary in magnitude (or existence) between genotypes; such variation thus constitutes a key area for further research in the field of female aggression.

We first test here whether there was an effect of male or female genotype on female aggression in three commonly used lab strains ($w^{1118}$, Canton S, and Dahomey), both before and after mating. We predicted that there would be differences between the genotypes in virgins' levels of aggression due to underlying genetic differences and to visual acuity ($w^{1118}$ flies are visually impaired [25]). We also predicted that differences amongst genotypes in males' ability to stimulate PMRs and differential female PMRs would lead to differences between genotypes in how much female aggression increases after mating.

Second, we tested the timing of mating-induced female aggression to identify putative mechanisms regulating it. We have shown previously that the transfer of sperm is necessary for females to increase aggression 24 hours after mating [11], which suggests that the timing of sperm transfer and sperm storage may be important mediators of the induction of female aggression. We had two predictions as to when female aggression should begin to increase:

1. Aggression should increase immediately after mating due to the stimulation of mating per se or the transfer of male ejaculate components (similar to the post-mating receptivity effect) OR;

2. Aggression should only increase a few hours after mating, when oviposition rates begin to increase, suggesting a direct link between female aggression and egg laying [26,27].

Finally, we evaluated how well existing automated tracking and classification software is able to track and score female aggression using our current set-up. Studying behaviors such as aggression can be time-consuming as individuals often spend a relatively small fraction of time engaging in aggressive encounters. The invention of software to track animal locomotion and behavior has dramatically improved our ability to study broad-scale and individual-level behaviors [28]. Machine-learning techniques can now recognize specific suites of behaviors automatically, while automated tracking and classification software remove the need to manually score behaviors–empowering us to conduct high-throughput behavioral analyses. The use of such software to study male aggression in *Drosophila melanogaster* has led to the discovery of sex-specific genes, hormones, and neuronal pathways responsible for male aggression in flies and other animals [16,29]. However, to our knowledge, no studies have yet used automated tracking and classification software to study female aggression in *D. melanogaster* or other insects. Given that modules of fruit fly aggression (e.g. headbutting) are discrete and distinct from one another, machine learning-based methods of study may also be particularly useful for behavioral analyses in the fruit fly [20]. Developing a system to accurately record, track, and score female aggression is therefore a key methodological advancement which will facilitate further study of female aggression.

## Materials and methods

### Fly stocks and culture

To test whether an increase in female aggression after mating is common across different genotypes, we used three common laboratory genotypes: Dahomey, Canton-S, and $w^{1118}$. We chose these three genotypes as they are widely used as a genetic background for introgressing different mutations into in a variety of laboratory studies [30]. Flies from the Dahomey stock population were collected from Dahomey, Benin in the 1970s and have been kept in large, outbred population cages in the laboratory ever since [31]. The Canton S stock was received from the Heisenberg lab at the University of Würzburg, while $w^{1118}$ came from the Wilson lab at the University of Oxford. Both Canton S and $w^{1118}$ stocks were kept in smaller populations, but raised in large population cages for multiple generations prior to this experiment. Eggs were collected from each population cage and larvae from all genotypes were raised at standardized larval density [32]. Flies were collected within 8 hours of eclosion to ensure virginity, and the sexes were housed separately. Flies were raised and maintained as adults on standard fly food medium (for 1L of fly food, main ingredients followed this ratio: 20g molasses, 14.6g yeast, 6.8g agar, 5.6mL propionic/phosphoric acid) without live yeast. Males were kept in vials in groups of 10, while females were kept in individual vials from the time of eclosion to the time

they were used in a contest (i.e. five days). All flies were kept and experiments conducted at 25˚C on a 12:12 dark: light cycle.

## Experimental design

**Genotype experiment.**   In addition to testing whether females differ in aggression due to genotype, we also tested whether males from different genotypes differed in how much aggression they stimulated in females after mating. In all previous studies, females were mated to males from the same genetic background as them–i.e. Canton-S females were mated to Canton-S males, while Dahomey females were mated to Dahomey males [11,20]. As males from different genotypes may differ in their ejaculate composition or ability to stimulate female aggression, we also incorporated male genotype into our experimental design. To investigate whether any differences in genotype were due to male or female genotype, or an interaction between the two, we used a fully factorial design. Males from all genotypes were mated to females of all genotypes (sample sizes available in S1 Table). The genotype experiment was conducted in two blocks.

**Timing experiment.**   To test when female aggression increases after mating, we tested females at 6 time points after mating:

- 1 hour post-mating

- 2 hours post-mating

- 4 hours post-mating

- 8 hours post-mating

- 24 hours post-mating

- 1 week (168 hours) post-mating

For this experiment, we used only Dahomey females, as the results from our first experiment suggested there were no differences between genotypes in mating-induced aggression. In the 'mated' treatments, females were mated to standard Dahomey males. Each mated female was paired with a mated female from the same treatment that had finished mating within 30 minutes of each other, so that they were as closely matched as possible for end of mating time. For each pair of mated females, a pair of randomly selected virgin females was chosen and fought at the same time to get an equivalent control for each time point (sample sizes in S2 Table).

## Behavioral experiments

In the genotype experiment, we painted females 3 days post-eclosion (one day prior to mating, two days prior to fighting). Females were painted with either a red or yellow dot of acrylic paint on their thorax to facilitate individual identification. Females were not painted in the timing experiment.

4 days post-eclosion, females in the 'mated' treatments of both experiments were placed into a vial with a single male and observed. Once a single mating occurred, they were separated from these males and put into a fresh vial containing regular fly food media. Females that did not mate within 5 hours were discarded. We recorded the latency to mating and the duration of each mating for all pairs.

In the genotype experiment, females remained in their vials for 24 hours after mating. These vials were subsequently frozen and we counted the number of eggs each female had laid. In the genotype experiment, females were used in contests 24 hours after mating (5 days post-

eclosion). For the two hours directly before being used in a contest, females were kept in a vial with damp cotton wool but no food to increase the chance that we would see aggressive behavior [19].

In the timing experiment, the amount of time females spent in starvation vials prior to being used in contests differed depending on their treatment. Females from the '1 hour' and '2 hour' treatments were placed immediately into starvation vials upon completion of mating. Females from the other treatments were placed into starvation vials 2 hours before being placed in the contest arena (i.e. for females in the '4 hour' treatment, they remained in vials with access to food for 2 hours and then were in starvation vials for 2 hours).

Flies were then aspirated from these vials into a contest arena (diameter 2 cm) containing an Eppendorf tube cap filled with regular fly food (diameter 5 mm) and a ~2-μl drop of yeast paste, providing a limited resource to fight over. Females were allowed 5 minutes to acclimatise to the arena and were then filmed for 30 minutes using Toshiba Camileo X400 video cameras.

In the genotype experiment, females were removed from the contest arena after being recorded and placed into new vials with fly food (again containing no live yeast) and left to lay eggs. 24 hours later, these females were removed and the vials frozen to count the number of eggs females laid in the 24 hours after the contest. We then measured wing area of females used in contests as a proxy of body size [33].

## Manual behavioral scoring

Videos were scored blind to treatment. Only headbutts were recorded as a proxy for female aggression, as these have previously been shown to be the most common high intensity form of aggression engaged in by females [20,21]. We recorded the number of headbutting encounters and duration of each encounter. An encounter began when one female headbutted the other and ended when one fly left the food cap, when the flies were one body length apart, or had stopped physically touching each other for three seconds (all measures of the end of an aggressive interaction) (NB: an encounter may include multiple headbutts from one or both individuals). We then used total encounter duration as the primary response variable, as we have previously shown that it encompasses variation in both the number and duration of encounters to give a good overall indicator of the amount of aggression shown by a pair [11]. We also analysed the number of encounters involving headbutts to provide an additional measure of aggression. Using headbutts also enabled us to have a direct comparison with the automated behavioral data, where headbutting is detectable, but not lower intensity behaviors, such as fencing.

## Automated tracking and analysis

To track the females, we used the Caltech fly tracker [34]. The program records the location and trajectory of individual flies, as well as producing data on other parameters, such as velocity, distance to other individuals, and location within an arena. These parameters and tracking data were then transferred to the program JAABA to use machine learning to automatically classify headbutting behavior in our videos [35]. We first calibrated the JAABA machine learning algorithm by manually annotating several video frames in a subset of videos to specify which behaviors are of interest–in our case labelling frames that contained instances of headbutting and a sample of those frames that did not contain headbutting. The program then conducted an iterative process of machine learning to identify further cases of that behavior seen in the videos by using the annotated frames as a reference point. These predictions were then

checked by the manual trainer, correcting and refining the program's predictions. Once further refinements to the algorithm generated no improvement in the program's ability to classify aggression (as measured using JAABA's ground-truthing function), we used this classifier across all videos in both the timing and genotypes experiment.

We could use the automated tracking and classification analysis software on 189 of the 227 videos that we were able to score manually in the timing experiment. In the genotype experiment, we were able to use tracking and classification software on 272/332 videos. The tracking software failed to successfully track the flies in the remaining videos in both experiments (potentially due to issues related to lighting and contrast levels).

## Statistical analysis

We performed all statistical analysis in R (version 3.3.2) [36]. In the timing experiment, for analyses involving contest duration and headbutt number we used generalized linear models with negative binomial distributions [using the function glm.nb from the 'MASS' package—[37], as these models best fit our data, and our data met the majority of the assumptions for such models. To compare the results of manual and automated scoring, we used a linear model with the manual scores as the response variable, with automated scores and mating status (and their interaction) as the explanatory variables.

For the genotype experiment, we used GLMs with Gamma distributions to analyze contest duration and headbutt number data, as these models fit our data better than linear models or GLMS with negative binomial distributions. We fitted two models:

1. To investigate whether males differed in their ability to stimulate female aggression (and whether there was an interaction between male and female genotype), we fit the following model on a dataset that contained only mated females:

$$\text{Contest duration} \sim \text{Male genotype} * \text{Female genotype}$$

2. To test whether female genotypes differed in the magnitude of their response to mating, we pooled all male mating treatments together (i.e. for Dah females, the 'mated' treatment consisted of females mated to Dah males, Canton-S males, and $w^{1118}$ males). We then fit the following model to a dataset containing all females:

$$\text{Contest duration} \sim \text{Mating status} * \text{Female genotype}$$

In addition, we investigated the effect of female genotype on wing area, and the number of eggs a female laid in the 24 hours before-, and after being used in a contest. For the wing area analysis, we used a linear model with wing area as the response and female genotype as the explanatory variable. For the egg count analyses, we fit a GLM with a quasipoisson distribution as the data was count data that was overdispersed and therefore did not fit a Poisson distribution. As body size has been shown to be linked to fecundity in *D. melanogaster*, we included it in the model as follows:

$$\text{Egg count(either pre- or post- contest)} \sim \text{Mating status} * \text{Female genotype} * \text{Wing area}$$

## Results

### Genotype experiment

**Males from different genotypes do not differ in their ability to stimulate female aggression.** Males of different genotypes did not differ in the aggression they stimulated in mated females ($Dev_{2, 178} = 0.12$, P = 0.87; S1 Fig). Female genotype was marginally non-significant ($Dev_{2, 176} = 2.26$, P = 0.071), with a trend for $w^{1118}$ females to fight for less time than Canton-S or Dahomey females. There was no significant interaction between male and female genotype ($Dev_{4, 172} = 2.62$, 0.19), which suggests that males do not stimulate a different amount of aggression in females of their own genotype compared with those from a different genotype. There was no significant effect of male genotype ($Dev_{2, 178} = 0.43$, P = 0.48), female genotype ($Dev_{2, 176} = 0.88$, P = 0.22), or their interaction on the number of headbutts females performed ($Dev_{4, 172} = 1.47$, P = 0.28).

**Mating-induced female aggression is consistent across different genotypes.** Mated females fought for longer than virgin females across all female genotypes (GLM with Gamma distribution: $Dev_{1, 270} = 6.78$, P = 0.0002; Fig 1). There was also a significant effect of genotype, with $w^{1118}$ females fighting for less time than either Canton-S or Dahomey females when both mated and virgin ($Dev_{2, 268} = 4.95$, P = 0.006). There was no significant interaction between mating status and female genotype, suggesting that the genotypes showed similar increases in aggression in response to mating ($Dev_{2, 266} = 1.37$, P = 0.247). Similarly, mated females performed more headbutts than virgin females ($Dev_{1, 266} = 4.1$, P = 0.0003), with $w^{1118}$ females performing fewer headbutts than either Dahomey or Canton-S ($Dev_{2, 266} = 3.14$, P = 0.007), with no significant interaction between mating status and female genotype ($Dev_{2, 266} = 1.26$, P = 0.14).

### Timing experiment

**Aggression increases 2–4 hours after mating.** There was a significant interaction between mating status and number of hours after mating for contest duration (GLM with negative binomial distribution: $Dev_{5, 213} = 25.23$, P = 0.0001; Fig 2a). There was no significant difference between mated and virgin females 1 or 2 hours after mating, but there were significant differences at 4 hrs, 8 hrs, 24 hrs, and a week after mating (GLMs conducted on each time point separately: 1 hr: $Dev_{1,31} = 0.66$, P = 0.42, 2 hrs: $Dev_{1,32} = 0.11$, P = 0.74, 4 hrs: $Dev_{1,43} = 24.81$, P < 0.001, 8 hrs: $Dev_{1,24} = 6.36$, P = 0.01, 24 hrs: $Dev_{1,41} = 24.33$, P < 0.001, Week: $Dev_{1,44} = 14.32$, P < 0.001). There was a significant main effect of mating, whereby mated females fought for longer than virgin females ($Dev_{1,225} = 52.877$, P < 0.0001). There was no significant effect of number of hours after mating or of block (Hours: $Dev_{5, 220} = 8.496$, P = 0.14; Block: $Dev_{2, 218} = 3.217$, P = 0.2). Similar results and patterns were obtained for headbutt number, with a significant interaction of mating and number of hours after mating ($Dev_{5, 213} = 28.57$, P < 0.0001), and a significant main effect of mating ($Dev_{1,225} = 56.44$, P < 0.0001), but no other effects were significant (Hours: $Dev_{5, 220} = 9.53$, P = 0.09; Block: $Dev_{2, 218} = 2.18$, P = 0.34).

**How effective is using automated tracking and machine learning software for studying female aggression?.** We tested whether there was a correlation between manual observations and tracking observations of headbutts, as well as whether this effect differed by mating status. We found a significant positive correlation between manual and tracking data (LM: $F_{1, 185} = 14.05$, P = 0.0002, Adjusted $R^2 = 0.21$; Fig 3), a significant effect of mating status for manual data (as found previously: $F_{1, 185} = 31.52$, P < 0.0001), and a significant interaction between mating status and tracking data ($F_{1, 185} = 6.04$, P = 0.015). As can be seen in Fig 3, the

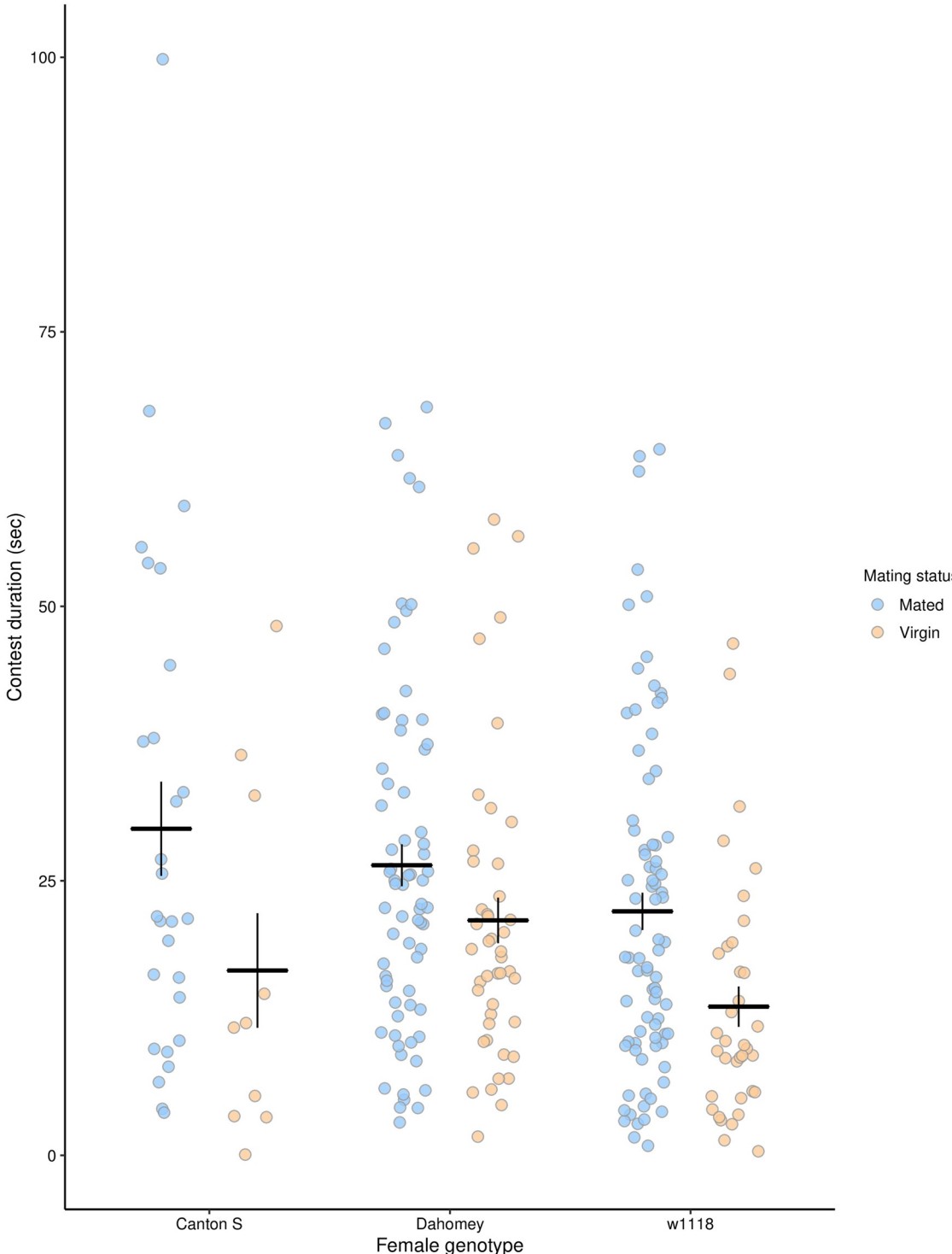

**Fig 1. Mated females fought for longer than virgin females in all genotypes and *w^1118* females fought for less time than Canton-S and Dahomey females.** Blue points represent mated females while yellow points represent virgin females in each genotype. Each point represents the contest duration for one pair of females. The black bars represent the treatment means ± 1 standard error. Sample sizes are recorded in S1 Table.

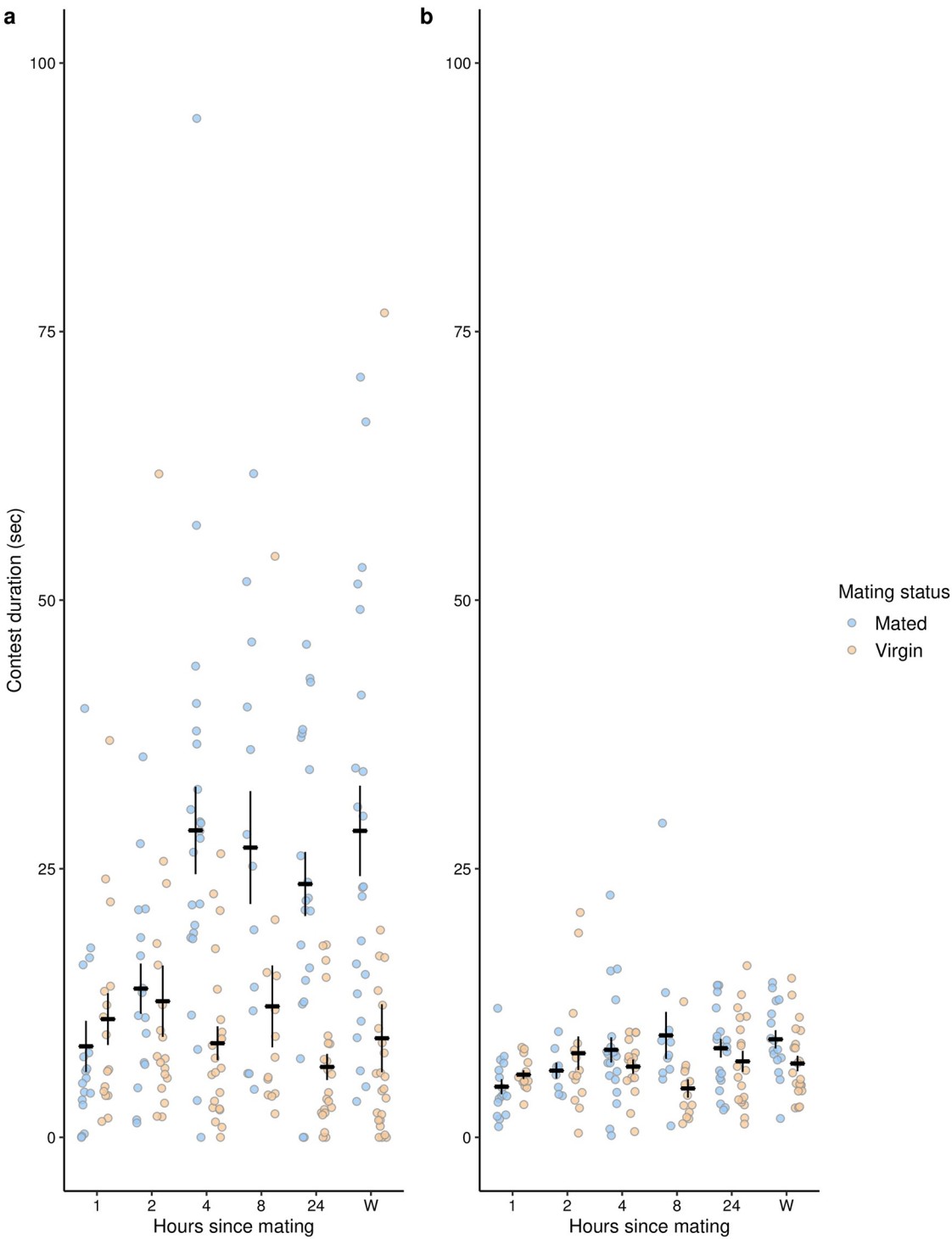

**Fig 2. Mated females start to fight for longer than virgins around four hours after mating.** a. Manual scoring of headbutt duration. b. Automated tracking and classification of headbutt duration. Blue points represent mated females, while yellow points represent virgin females. Each point represents the contest duration for one pair of females. The black bars represent the treatment means ± 1 standard error. The points in a and b represent the same videos scored manually (in a) and by the automated system (in b). We place them side by side here for comparison of the different methods.

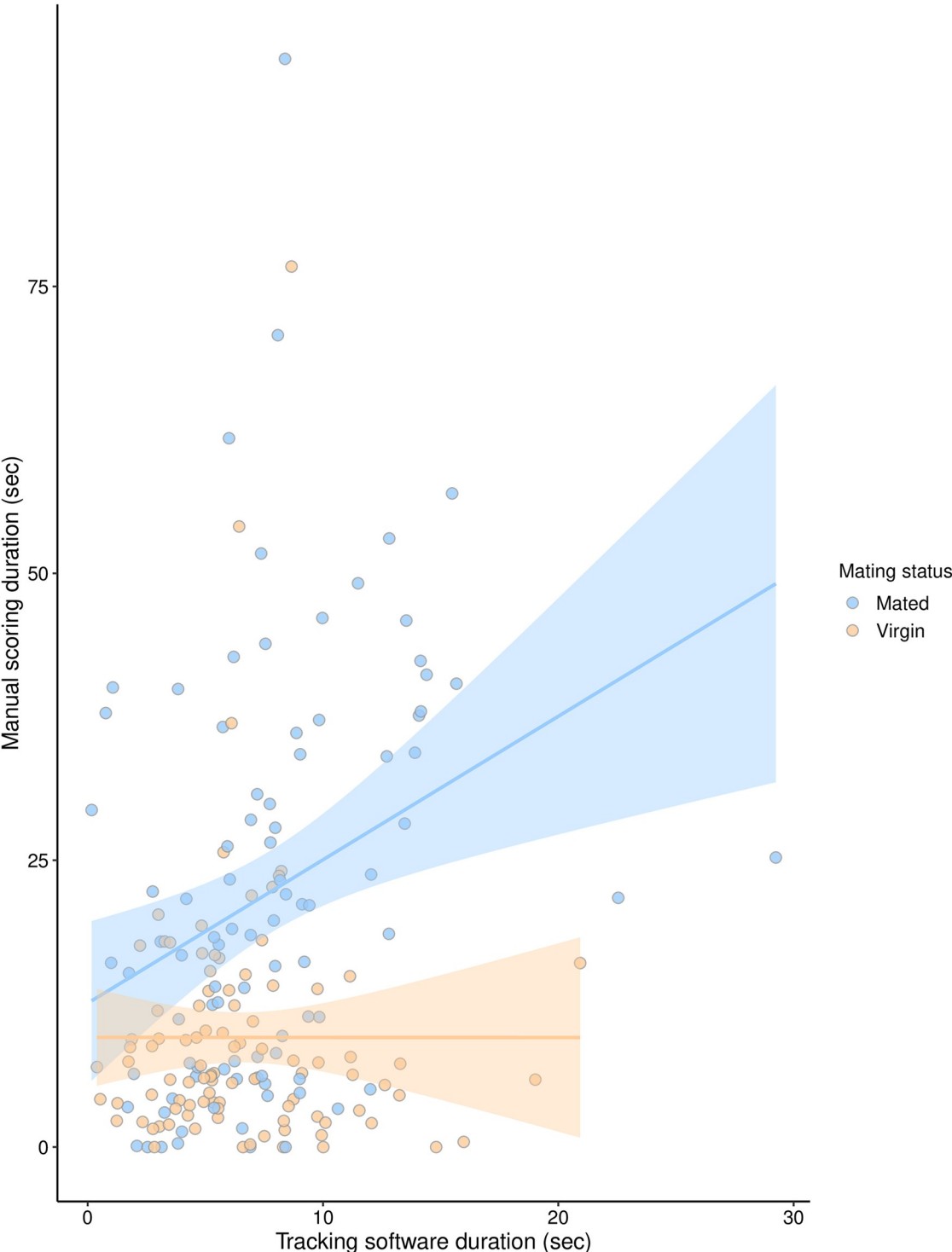

**Fig 3. Manual and tracking data are positively correlated for mated females but not for virgin females.** Blue points represent mated females from all treatments, while yellow points represent virgin females from all treatments. Each point represents the contest duration for one pair of females. The lines indicate the linear model fit between the manual scoring and tracking data for mated (blue) and virgin (yellow) females separately. The highlighted areas around the line indicate the 95% confidence interval.

correlation between the manual and tracking data was significantly positive for mated females (linear regression equation: Y = 12.54 + 1.25x, P = 0.0002), while the slope for virgin females was not significantly different from 0 (Y = 9.56–0.0005x, P = 0.99).

## Discussion

### Genotypes

We found that mating stimulated female aggression in all three genotypes to a similar degree and that males from different genotypes did not differ in their ability to stimulate aggression. Our results suggest that mating-induced female aggression is common to multiple genotypes of *Drosophila melanogaster* and is present in similar magnitudes.

Genetic variation underlying female aggression is not well-studied across taxa. While sex-specific genetic architectures have been investigated for a number of life history and morphological traits [38,39], there have been fewer studies on behavioral traits [40]. Behavioral traits are often highly plastic and depend on an individual's social and physical environment, which may suggest a relatively small genetic component responsible for variation in female aggression. Shorter et al (2015) tested male aggression in 200 inbred lines of *Drosophila melanogaster* to detect genetic variation underlying aggression. They found a 20-fold difference between the most and least aggressive lines, showing a strong effect of genotype on male aggressive behavior. Although this study found high levels of broad-sense heritability ($H^2 = 0.69 \pm 0.07$), they found very low narrow-sense heritability ($h^2 = 0.00$), which suggests a complicated set of gene interactions determining male aggressive phenotypes [29].

Male aggression in *Drosophila melanogaster* has been shown to be regulated by a sex-specific neural pathway, which suggests that there may be very different genetic architectures for male and female aggression in this species [41]. We found that there was a genotype effect on female aggression (for both mated and virgin females), whereby $w^{1118}$ females fought for less time than Canton-S or Dahomey females. These results could suggest genotypic differences underlying aggression, but it is also possible that it due to a phenotypic manifestation of the white-eye mutation in $w^{1118}$ females. Flies with a white-eye mutation are visually impaired, which detrimentally affects their locomotion and courtship behavior during photophase [42,43]. $w^{1118}$ females fighting for less time is consistent with a specific effect of the white-eyed mutation, rather than relying on broader polygenic differences between genotypes. We also found that $w^{1118}$ males were much slower to mate than either Canton-S or Dahomey males and mated for less time (S1 File and S2 & S3 Figs), as expected given their reduced ability to locate and court females [44]. Our results suggest that there could be underlying genetic differences related to female aggression, but there do not appear to be dramatic differences between our female genotypes that alter either their base level of aggression or their aggressive response to mating.

Females from different populations can vary significantly in their response to mating and male ejaculates [24], while males can also differ greatly in their ability to stimulate post-mating responses based on their condition and previous social environment [45–47]. However, we did not find any effect of male genotype or its interaction with female genotype on inducing female aggression or for either of our egg production measures (S1 File and S4 & S5 Figs). Our results suggest that males from these three genotypes may not differ significantly in their ejaculate, or at least in those components that influence female aggression and egg laying in the 48 hours after mating, or that the post-mating female aggression response is robust to variation in ejaculate composition. There is previous evidence, however, that shows that female aggression can vary in magnitude based on qualities of the male ejaculate. Females raised at different larval densities showed different levels of increase in aggression after mating [48]. Females raised

at higher larval density showed a greater increase in aggression after mating, which is probably due to their small body size meaning that the ejaculate they receive from males is larger in proportion to their mass [47].

## Timing

We found that female aggression in *Drosophila melanogaster* increased in mated females between two and four hours after mating, and remained elevated for at least a week after a single mating. There did not appear to be an increase over time in the level of aggression after mating–once females displayed an increase in aggression (2–3.5-fold higher than virgins), the difference between mated and virgins seemed to remain consistent for at least a week. This consistency, combined with the fact that males of different genotypes did not stimulate different levels of aggression, suggests a potential 'switch'-like mechanism for female aggression–i.e. there is no gradual build-up of aggressive behavior, but instead once aggression has been turned on, it remains on at the same level. Ovipositor extrusion has been suggested to be such a behavior–it is only present in mated females, not in virgin females [49]. Other behavioral and physiological effects show more of a gradual build-up effect, turning certain pathways present in virgins up (or down) in mated females, such as sex peptide increasing oogenesis and reducing receptivity in mated females [50]. It is possible that intermediate levels of aggression may produce few benefits but still display the same costs associated with expressing higher levels of aggression, suggesting an 'on/off' switch may be a more beneficial way to regulate aggression.

One possible interpretation of the switch-like nature of female aggression is that increased aggression after mating is an adaptive response by females. It may be beneficial for females to only upregulate their aggression after mating as only then is it necessary for them to compete over access to resources such as food or oviposition sites. As virgins, their rate of egg production is low, as is their need for protein-rich foods, which suggests that the benefits to competing over such resources are limited. Potentially, when females deplete their sperm stores we may expect a return to virgin-like aggression, although this remains unknown. As yet, we have shown no direct costs to aggression in females, but it seems likely that there are at least some energetic costs to engaging in aggressive encounters with other females in other taxa [12]. Taken together, these suggest a low cost-benefit ratio for virgin females for engaging in aggression, but this may shift for mated females, leading to increased aggression. Females across taxa are quite plastic in their expression of aggression, with some authors suggesting that females should be even more plastic than males due to the potentially higher costs from engaging in aggressive encounters [3]. Overall, there seems to be clear evidence that females can adjust their levels of aggression in response to their environment or reproductive status, and our study may represent another instance of females altering their aggression in such an adaptive fashion.

**Breakdown of timing in mated females.** To identify the putative mechanisms regulating female aggression after mating, it is useful to consider the timing of other post-mating responses (Fig 4). Matings generally take around 15–20 minutes in *Drosophila melanogaster*, with sperm transfer taking 1–8 minutes and the remaining duration taken up with the transfer of Sfps [51]. The timing of sperm transfer and sperm storage may be important mediators of the induction of female aggression as the transfer of sperm is necessary for females to increase aggression 24 hours after mating [11]. Sperm storage begins around 25 minutes after the start of mating, with females storing up to 400 sperm in their seminal receptacle by 1 hour after the start of mating [27]. Females then use these sperm to fertilize their eggs over the next 5–7 days, depleting their sperm store [27,52]. Female aggression may be triggered by females reaching

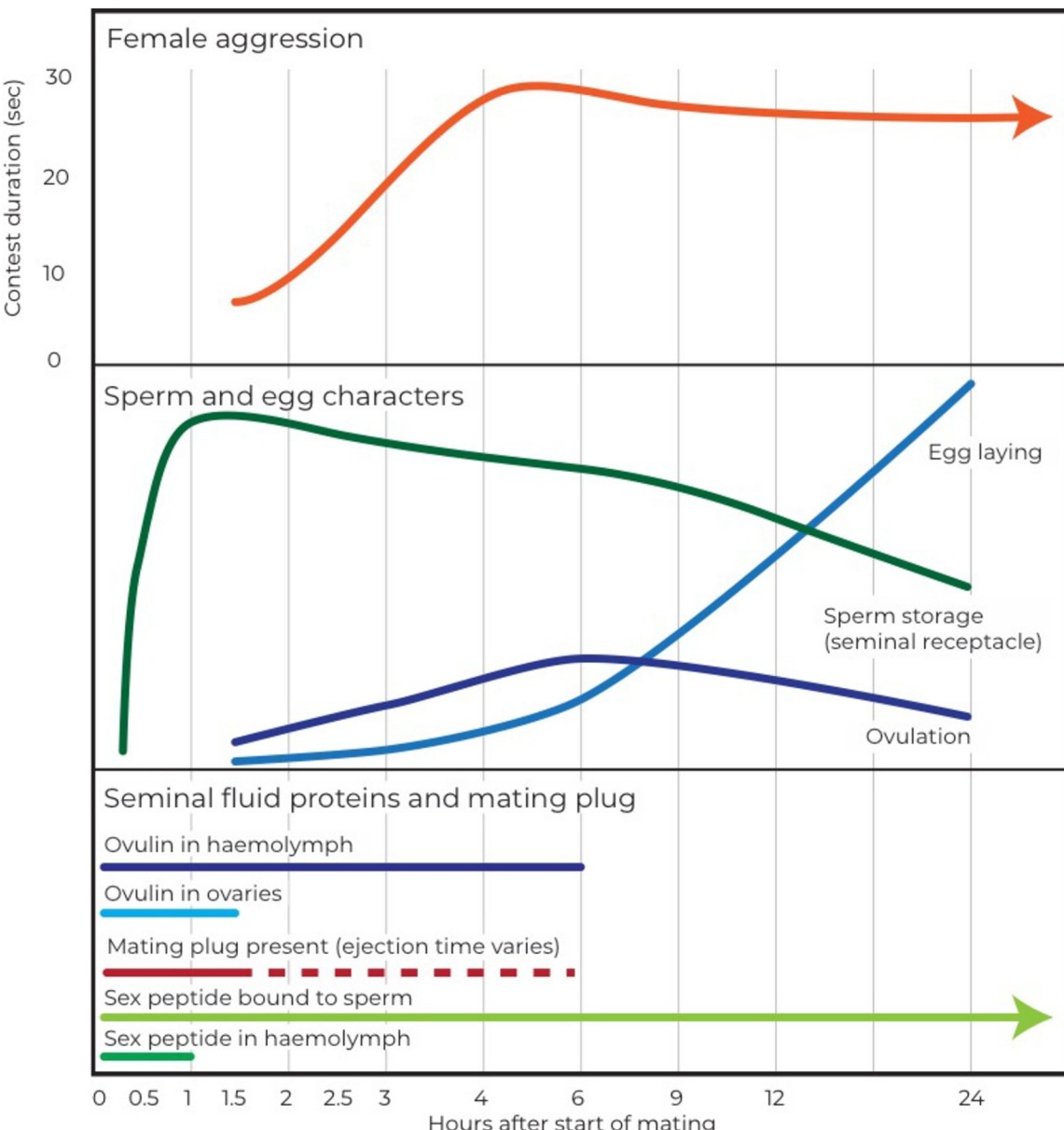

**Fig 4. Schematic of timing of various processes in mated females for the first 24 hours after mating.** A. The top portion of the figure represents aggression in mated females over the first 24 hours after mating as contest duration in seconds. This matches Fig 1 in this paper. B. The middle part of the figure demonstrates the patterns of sperm storage (in the seminal receptacle only), egg laying (oviposition), and ovulation in mated females. These lines represent general patterns and are not to scale with each other, but merely to give an indication of when and how these processes change over time. Sources: sperm storage [27], ovulation [26], egg laying [26]. C. The bottom portion of the figure demonstrates the timing of the presence of various seminal fluid proteins in different parts of the female–e.g. the top line represents when ovulin has been detected in the female hemolymph after mating. Sources: ovulin in hemolymph [53,54], ovulin in ovaries [26], mating plug and sperm ejection [23,55], sex peptide bound to sperm [56], sex peptide in hemolymph [57].

their maximum sperm storage capacity, which appears to occur before the onset of increased aggression in mated females (Fig 4). Female aggression does not appear to follow the same pattern as sperm number—sperm stores usually diminish over a period of time (dependent on female fecundity and number of sperm stored initially—[27,52]), whereas female aggression remains at an elevated level from 4 hours to a week after mating. This could suggest that the

'off' part of the female aggression switch is not determined by the number of sperm in storage. However, we did not test the number of sperm females had at different time points, so cannot be sure that our females showed the same pattern of sperm depletion as other females. Our females were kept without live yeast, which results in fewer eggs being laid and therefore fewer sperm being used, so they may have retained high numbers of sperm.

Other than sperm, seminal fluid proteins may also be involved in inducing post-mating aggression in females. 'Sex peptide' (SP) is at least partially responsible for the increase in aggression after mating and acts at multiple timescales, from a few hours after mating to a week after mating [56,58,59]. These timescales line up with our findings in this study–that female aggression increases 2–4 hours after mating and remains elevated for at least a week.

The increase in female aggression 2–4 hours after mating also appears to occur in conjunction with the stimulation of ovulation (1.5 hours after mating starts) and potentially also with the beginning of egg laying (~3 hours after the start of mating) [26,60]. We have previously shown that the production of eggs is not necessary for mating-induced female aggression and our results from both the timing and genotype experiments lend support to this idea [11,48].

## The use of automated tracking and classification software

Interestingly, there was a significant difference in how closely matched the automated and tracking data were between mated and virgin pairs. For mated females, videos that had higher scores from manual scoring also generally had higher scores when tracked. The fitted regression line for mated females had a slope of 1.25, suggesting that the tracking data may show lower values for dyads with less fighting than manual scoring and higher values for dyads with more fighting than manual scoring. Although there was a lot of variability in the tracking-manual data relationship, it seems tracking may give a relatively accurate indication of aggression for mated females. In contrast, the manual and tracking scores showed very little concordance for virgin females (slope of fitted line = 0.0005, Fig 3). There were generally fewer virgin pairs with higher scores (as virgin females fought less overall), but we would still expect to see a positive correlation between manual and tracking data over a narrower range of values.

This difference in the ability of the tracking system to detect aggression in mated and virgin females may indicate that there are not only quantitative differences between the two, but also qualitative differences. In a detailed ethographic comparison of female aggression, [20] found only small differences in fighting behavior between mated and virgin females. They found that mated females performed more behavioral transitions than virgins between different types of aggressive behaviors–e.g. from different types of fencing to headbutting to wing threats. However, they did not report any differences in the nature of individual behaviors, such as differences in how fast females performed headbutts, which our study potentially provides some evidence for.

We report the particular issues and challenges we faced using automated tracking and classification software for female aggression in the Supplementary Information as we hope this will assist others when designing their own setups. If tracking and classification software could be made to work for the relatively subtle behaviors of female aggression, it would dramatically reduce the amount of time required for each experiment. Currently, manually scoring videos can take weeks or months of valuable researcher time and restricts the number of treatments and replicates that can be used in any study on female aggression.

## Conclusions

Increasing our understanding of variation in female aggression is crucial for our comprehension of its development, function, and evolutionary consequences. Given the relationship

between aggression and fitness is likely to differ substantially between males and females, it is vital to consider female-specific morphology, physiology, and behavior to fully understand the evolution of female aggression across species. Many proximate mechanisms of male aggression have been shown to be conserved across taxa—the neurotransmitter serotonin is found to play a role in both invertebrate and vertebrate male aggression, suggesting one of its very early functions may have been to regulate male aggression [61,62]. It is possible that there are similar mechanisms that regulate female aggression that have yet to be properly examined–candidates include juvenile hormone in insects, and testosterone in vertebrates [15,63]. Studying the temporal and genetic variation of female aggression allows us to identify putative proximate mechanisms and further test their role in the development and evolution of female aggression.

## Supporting information

**S1 Fig. Mated females fight for longer than virgin females but this does not depend on the genotype of their mate.** Colours indicate the genotype of the male that a female mated with– blue = Canton-S, yellow = Dahomey, red = $w^{1118}$, empty circle = virgin female. Black bars indicate treatment means ± 1 standard error.
(DOCX)

**S2 Fig. Female genotypes did not differ in their mating duration, but $w^{1118}$ males mated for less time than Canton S and Dahomey males.** Colours indicate the genotype of the male that a female mated with–blue = Canton-S, yellow = Dahomey, red = $w^{1118}$. Black bars indicate treatment means ± 1 standard error.
(DOCX)

**S3 Fig. $w^{1118}$ males took longer to start mating, while Canton S females started mating the fastest.** Colours indicate the genotype of the male that a female mated with–blue = Canton-S, yellow = Dahomey, red = $w^{1118}$. Black bars indicate treatment means ± 1 standard error.
(DOCX)

**S4 Fig. $w^{1118}$ females had smaller wings than Dahomey and Canton S.** Colours indicate the genotype of the female–blue = Canton-S, yellow = Dahomey, red = $w^{1118}$. Black bars indicate treatment means ± 1 standard error. All females used in contests were included in this figure.
(DOCX)

**S5 Fig. Egg counts from 24 hours prior to contests (0–24 hours post-mating).** Colours indicate the genotype of the male that a female mated with–blue = Canton-S, yellow = Dahomey, red = $w^{1118}$, empty circle = virgin female. Black bars indicate treatment means ± 1 standard error.
(DOCX)

**S6 Fig. Egg counts for 24 hours post-contest (48–72 hours post-mating).** Colours indicate the genotype of the male that a female mated with–blue = Canton-S, yellow = Dahomey, red = $w^{1118}$, empty circle = virgin female. Black bars indicate treatment means ± 1 standard error.
(DOCX)

**S7 Fig. Contest arena setup.** This figure shows the setup we used for contests between females, which consisted of a circular arena with a food cap set into the middle. This food cap contained regular fly food medium, with a drop of yeast paste in the middle to act as a valuable, restricted resource. In this image, you can see a shadow that covers the top portion of the arena, which influences the reliability of the tracking software. You can also see the restricted contrast between flies on the food cap compared with flies on the white background, which again

makes it difficult for the tracking software to detect the flies accurately. The label at the top of the screen indicates the treatment, which was covered during observations and manual scoring of behaviors.
(DOCX)

**S1 Table. Sample sizes for genotype experiment.** Sample sizes indicate the number of pairs that the tracking software successfully tracked for each treatment in the genotypes experiment.
(DOCX)

**S2 Table. Sample sizes for timing experiment.** Sample sizes indicate number of pairs in each treatment. The number of pairs that were scored manually (the total number of pairs for that treatment) is recorded in the first column, while the number of pairs that were scored using automated tracking and analysis software are in the second column for both mated and virgin pairs.
(DOCX)

**S1 File.**
(DOCX)

# Author Contributions

**Conceptualization:** Eleanor Bath, Stuart Wigby.

**Data curation:** Eleanor Bath.

**Formal analysis:** Eleanor Bath.

**Funding acquisition:** Eleanor Bath, Stuart Wigby.

**Investigation:** Eleanor Bath, Edmund Ryan Biscocho, August Easton-Calabria.

**Methodology:** Eleanor Bath, Edmund Ryan Biscocho, August Easton-Calabria.

**Project administration:** Eleanor Bath.

**Resources:** Eleanor Bath, Stuart Wigby.

**Software:** Eleanor Bath.

**Supervision:** Eleanor Bath, Stuart Wigby.

**Validation:** Eleanor Bath.

**Visualization:** Eleanor Bath.

**Writing – original draft:** Eleanor Bath.

**Writing – review & editing:** Eleanor Bath, Edmund Ryan Biscocho, August Easton-Calabria, Stuart Wigby.

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
