## [Decision Letter · Decision Letter 0]

4 Mar 2020

PONE-D-20-03858

Temporal and genetic variation in female aggression after mating

PLOS ONE

Dear Dr Bath,

Thank you for submitting your manuscript to PLOS ONE. After careful consideration, we feel that it has merit but does not fully meet PLOS ONE’s publication criteria as it currently stands. Therefore, we invite you to submit a revised version of the manuscript that addresses the points raised during the review process.

Please respond thoroughly to the comments below.  Regarding the w1118 points, to the best of my understanding  visual impairment in these flies appears to be age dependent, but please respond to these comments as well.

We would appreciate receiving your revised manuscript by Apr 18 2020 11:59PM. To enhance the reproducibility of your results, we recommend that if applicable you deposit your laboratory protocols in protocols.io, where a protocol can be assigned its own identifier (DOI) such that it can be cited independently in the future. For instructions see: http://journals.plos.org/plosone/s/submission-guidelines#loc-laboratory-protocols

We look forward to receiving your revised manuscript.

Kind regards,

Efthimios M. C. Skoulakis, PhD

Academic Editor

PLOS ONE

Journal Requirements:

Reviewers' comments:

Reviewer's Responses to Questions

**Comments to the Author**

1. Is the manuscript technically sound, and do the data support the conclusions?

Reviewer #1: Partly

Reviewer #2: Yes

2. Has the statistical analysis been performed appropriately and rigorously? 

Reviewer #1: Yes

Reviewer #2: I Don't Know

3. Have the authors made all data underlying the findings in their manuscript fully available?

Reviewer #1: Yes

Reviewer #2: Yes

4. Is the manuscript presented in an intelligible fashion and written in standard English?

Reviewer #1: Yes

Reviewer #2: Yes

5. Review Comments to the Author

Reviewer #1: This manuscript by Bath et al., reports on the temporal variation of female aggression after mating in Drosophila melanogaster. Aggression, as a behavior itself and as driven by evolutionary mechanisms, is of considerably interest to many in the behavioral neuroscience community. The paper is very well written with an informative introduction and clear hypotheses and outcomes. The authors take care in describing both the strengths and weaknesses of their automated system. The conclusion that Drosophila female aggression increased in females between two and four hours after mating and remain elevated for at least a week is very interesting. With these results and recent studies including Wang et al., Nature Feb 2020, separate experiments examining neurons and circuits responsible for the increase in encounter duration upon mating should be doable and exciting. However, resolution of the following points are required.

Major points:

1. The number of matings and fights that have been generated for this manuscript is much appreciated, however, the sole reliance on contest duration weakens the ability to draw conclusions. The authors need to provide headbutt number comparisons as well. This should be in the contest duration anyway. Encounter duration is fine but there is an element of subjectivity in this calculation – when is the cut-off that females have “stopped interacting”? The females may be across from each other eating the yeast paste without a true separation but also not aggressively interacting. Providing the simple headbutt number and statistical analysis of this number needs to be included.

2. w1118 males have not been used to the best of my knowledge since Hoyer et al., 2008. This is because as described in this paper, “the eyes are visually impaired”, [refs within, Hengstenberg and Gotz, 1967, Wehner et al., 1967]. Currently, the background of many transgenic males are “cantonized” to eliminate this situation. The authors should either remove the w1118 data or explain that these females are visually impaired.

3. What Fig. 2b represents isn’t clear to me and the only mention in the text is line 335. Please explain – is this headbutt only data?

3. Maybe I missed this but what is the size of the contest arena? This is necessary for others to replicate the assay.

Minor points:

1. Reference formatting on lines 52-54

2. It would help to provide examples of when the authors hypothesize aggression will be the most beneficial to females, line 74.

3. Drosophila has a powerful model organism for the study of male aggression for decades and more than one reference should be given, line 81.

4. w1118 should be italicized (white gene) and 1118 should be superscript as it is the allele of this gene.

5. Line 133, lunging is a male aggressive pattern, it makes sense to use a female pattern here.

6. Line 135, automated analysis of male aggression has been described for years, do the authors mean female here?

7. If the authors software can detect differences in the speed of virgin vs mated headbutts that would be amazing. Headbutts are very fast and there is a second fly involved as with male lunging giving each pair pattern a level of uniqueness.

8. Is there a reason Fig. 4 is described in the Discussion and not in the Results section?

Reviewer #2: Bath et al. show that female aggression over food (measured as duration of headbutt behavior) increases following mating and persists for at least a week. This increase is robust and not dependent on the genotype of the female, or the genotype of the male she mates with. In addition, the authors have developed an automated tracking system and discuss its performance and challenges.

The authors articulate clear hypotheses and address them with carefully designed and executed experiments that are rigorously evaluated. The data support the conclusions the authors draw.

The discussion of the newly developed tracking system and some of its weaknesses is careful and will be of great value for further optimization of the automated approach.

6. PLOS authors have the option to publish the peer review history of their article (what does this mean?). If published, this will include your full peer review and any attached files.

Reviewer #1: No

Reviewer #2: No

---

## [Author Response · Author response to Decision Letter 0]

19 Mar 2020

We have attached a file ('Response to Reviewers') with our detailed responses to the comments from the reviewers and editors.

---

## [Decision Letter · Decision Letter 1]

1 Apr 2020

Temporal and genetic variation in female aggression after mating

PONE-D-20-03858R1

Dear Dr. Bath,

We are pleased to inform you that your manuscript has been judged scientifically suitable for publication and will be formally accepted for publication once it complies with all outstanding technical requirements.

With kind regards,

Efthimios M. C. Skoulakis, PhD

Academic Editor

PLOS ONE

Additional Editor Comments (optional):

Reviewers' comments:

Reviewer's Responses to Questions

**Comments to the Author**

1. If the authors have adequately addressed your comments raised in a previous round of review and you feel that this manuscript is now acceptable for publication, you may indicate that here to bypass the “Comments to the Author” section, enter your conflict of interest statement in the “Confidential to Editor” section, and submit your "Accept" recommendation.

Reviewer #1: (No Response)

2. Is the manuscript technically sound, and do the data support the conclusions?

Reviewer #1: Yes

3. Has the statistical analysis been performed appropriately and rigorously? 

Reviewer #1: Yes

4. Have the authors made all data underlying the findings in their manuscript fully available?

Reviewer #1: Yes

5. Is the manuscript presented in an intelligible fashion and written in standard English?

Reviewer #1: Yes

6. Review Comments to the Author

Reviewer #1: The authors have addressed the major and minor points that were raised.

In particular, I asked for headbutt numbers as the first submission relied solely on contest duration. The authors did provide the information and wrote that their previous work indicated contest duration and headbutt number are related. This is useful information which I may have missed in the first submission, however, it is better to not rely that any reader of this current manuscript has read their 2017 paper. Also, the authors are right that although the addition of the headbutt data did not change the interpretations, it does always strengthen a paper to provide more than one measure of aggression. Adding a supplemental figure or two for the headbutt data would have been useful, however the data in the text is acceptable.

7. PLOS authors have the option to publish the peer review history of their article (what does this mean?). If published, this will include your full peer review and any attached files.

Reviewer #1: No

---

## [Editor Report · Acceptance letter]

17 Apr 2020

PONE-D-20-03858R1 

Temporal and genetic variation in female aggression after mating 

Dear Dr. Bath:

I am pleased to inform you that your manuscript has been deemed suitable for publication in PLOS ONE. Congratulations! Your manuscript is now with our production department. 

With kind regards,

on behalf of

Dr. Efthimios M. C. Skoulakis 

Academic Editor

PLOS ONE